# The nucleotide pGpp acts as a third alarmone in *Bacillus*, with functions distinct from those of (p)ppGpp

Jin Yang [1], Brent W. Anderson [1], Asan Turdiev[2], Husan Turdiev[2], David M. Stevenson[1], Daniel Amador-Noguez [1], Vincent T. Lee [2✉] & Jue D. Wang [1✉]

The alarmone nucleotides guanosine tetraphosphate and pentaphosphate, commonly referred to as (p)ppGpp, regulate bacterial responses to nutritional and other stresses. There is evidence for potential existence of a third alarmone, guanosine-5′-monophosphate-3′-diphosphate (pGpp), with less-clear functions. Here, we demonstrate the presence of pGpp in bacterial cells, and perform a comprehensive screening to identify proteins that interact respectively with pGpp, ppGpp and pppGpp in *Bacillus* species. Both ppGpp and pppGpp interact with proteins involved in inhibition of purine nucleotide biosynthesis and with GTPases that control ribosome assembly or activity. By contrast, pGpp interacts with purine biosynthesis proteins but not with the GTPases. In addition, we show that hydrolase NahA (also known as YvcI) efficiently produces pGpp by hydrolyzing (p)ppGpp, thus modulating alarmone composition and function. Deletion of *nahA* leads to reduction of pGpp levels, increased (p)ppGpp levels, slower growth recovery from nutrient downshift, and loss of competitive fitness. Our results support the existence and physiological relevance of pGpp as a third alarmone, with functions that can be distinct from those of (p)ppGpp.

[1] Department of Bacteriology, University of Wisconsin, Madison, WI 53706, USA. [2] Department of Cell Biology and Molecular Genetics, University of Maryland, College Park, MD, USA. ✉email: vtlee@umd.edu; wang@bact.wisc.edu

Organisms from bacteria to humans rely on timely and appropriate responses to survive various environmental challenges. The stress signaling nucleotides guanosine tetraphosphate (ppGpp) and guanosine pentaphosphate (pppGpp) are conserved across bacterial species. When induced upon starvation and other stresses, they mediate multiple regulations and pathogenesis by dramatically remodeling the transcriptome, proteome, and metabolome of bacteria in a rapid and consistent manner[1–3]. (p)ppGpp interacts with diverse targets including RNA polymerases in *Escherichia coli*[4–8], replication enzyme primase in *Bacillus subtilis*[9–11], purine nucleotide biosynthesis enzymes[12–15], and GTPases involved in ribosome assembly[16–19]. Identification of (p)ppGpp-binding targets on a proteome-wide scale is one way to unravel a more extensive regulatory network[15,18,20]. However, because binding targets differ between different species and most interactomes have not been characterized, the conserved and diversifying features of these interactomes remain incompletely understood.

Another understudied aspect of (p)ppGpp regulation is whether ppGpp and pppGpp, while commonly referred to and characterized as a single species, target the same or different cellular pathways[21]. In addition, there is evidence for potential existence of a third alarmone, guanosine-5′-monophosphate-3′-diphosphate (pGpp), since several small alarmone synthetases can synthesize pGpp in vitro[22,23]. However, the clear demonstration of pGpp in bacterial cells has been challenging. More importantly, the regulation specificities and physiological importance of having multiple closely related alarmones in bacteria have not been systematically investigated.

Here we demonstrate pGpp as a third alarmone in Gram-positive bacteria by establishing its presence in cells, systematically identifying its interacting targets, and revealing a key enzyme for pGpp production through hydrolyzing (p)ppGpp. We also compare the targets of pGpp, ppGpp, and pppGpp through proteomic screens in *Bacillus anthracis*. We found that both pppGpp and ppGpp regulate two major cellular pathways: purine synthesis and ribosome biogenesis. In contrast, pGpp strongly regulates purine synthesis targets but does not regulate ribosome biogenesis targets, indicating a separation of regulatory function for these alarmones. In *B. subtilis* and *B. anthracis*, pGpp is efficiently produced from pppGpp and ppGpp by the Nucleoside Diphosphate linked to any moiety "X" (NuDiX) NuDiX alarmone hydrolase A (hydrolase NahA), both in vitro and in vivo. A Δ*nahA* mutant has significantly stronger accumulation of pppGpp and decreased accumulation of pGpp, as well as slower recovery from stationary phase and reduced competitive fitness against wild-type cells. Our work suggests a mechanism for the conversion and fine tuning of alarmone regulation and the physiological production of the alarmone pGpp.

## Results

### Proteome-wide screen for binding targets of pppGpp and ppGpp from *Bacillus anthracis*.
To systematically characterize the binding targets of (p)ppGpp and identify novel (p)ppGpp-binding proteins in *Bacillus* species, we screened an open reading frame (ORF) library of 5341 ORFs from the pathogen *Bacillus anthracis* (Fig. 1a). Using Gateway cloning, we placed each ORF into two expression constructs, one expressing the ORF with an N-terminal histidine (His) tag and the other with an N-terminal histidine maltose binding protein (HisMBP) tag.

We first characterized the binding targets of ppGpp using the *B. anthracis* library. To this end, each ORF in the HisMBP-tagged library was overexpressed and binding to [5′-α-$^{32}$P]-ppGpp was assayed using differential radial capillary action of ligand assay (DRaCALA)[24] (Fig. 1a). The fraction of ligand bound to protein

in each lysate was normalized as a Z-score of each plate to reduce the influence of plate-to-plate variation (Supplementary Data 1). We found that the strongest ppGpp-binding targets in *B. anthracis* can be categorized to three groups: (1) purine nucleotide synthesis proteins (Hpt1, Xpt, Gmk, GuaC, PurA, and PurR); (2) ribosome and translation regulatory GTPases (HflX, Der, Obg, RbgA, TrmE, and Era); and (3) nucleotide hydrolytic enzymes, including NuDiX hydrolases and nucleotidases (Fig. 1b). We compared these targets to those obtained from previous screens for ppGpp targets in *E. coli* and for an unseparated mix of pppGpp and ppGpp in *S. aureus*[18]. Comparison of our results with these previous screens yielded conserved themes (Fig. 1b). Among the most conserved themes are the purine nucleotide synthesis proteins (Fig. 1c) and ribosome and translation regulation GTPases (Fig. 1d).

Next, we performed a separate screen to characterize the binding of the *B. anthracis* proteome to pppGpp (Fig. 1a). pppGpp is the predominant alarmone induced upon amino acid starvation in *Bacillus* species, rising to a higher level than ppGpp. However, despite potential differences in specificity between pppGpp and ppGpp, the pppGpp interactome has not been systematically characterized in bacteria. We used both His-tagged and HisMBP-tagged libraries to test pppGpp binding. Using two differentially tagged libraries allows us to identify more potential hits and minimize false negatives. We found that pppGpp shares almost identical targets with ppGpp, with similar or reduced binding efficacy for most of its targets compared to ppGpp (Supplementary Data 1). By sharing targets with ppGpp, pppGpp also comprehensively regulates purine synthesis and ribosome assembly. We also found that several proteins bind to pppGpp but not ppGpp, including the small alarmone synthetase YjbM (SAS1). This is expected for YjbM, since it is allosterically activated by pppGpp, but not ppGpp[25].

### NahA, a NuDiX hydrolase among the (p)ppGpp interactome in *Bacillus*, hydrolyzes (p)ppGpp to produce pGpp in vitro.
The putative NuDiX hydrolase, BA5385, was identified as a novel binding target of (p)ppGpp. Protein sequence alignment showed that BA5385 has homologs in different *Bacillus* species with extensive homology and a highly conserved NuDiX box (Fig. S1). We cloned its homolog, YvcI, from the related species *Bacillus subtilis* and showed that overexpressed *B. subtilis* YvcI in cell lysate also binds ppGpp and pppGpp (Fig. 2a). The binding is highly specific, as non-radiolabeled ppGpp effectively competes with radiolabeled (p)ppGpp binding, whereas non-radiolabeled GTP failed to compete. EDTA eradicated (p)ppGpp binding to His-MBP-YvcI cell lysate, which implies that the divalent cation present in the reaction (Mg$^{2+}$) is essential for (p)ppGpp binding to YvcI (Fig. 2a).

We noticed that while YvcI-overexpression cell lysate showed strong and specific binding to (p)ppGpp, the purified protein does not appear to bind (p)ppGpp in DRaCALA (Fig. S2). This suggests that either YvcI requires a co-factor present in the lysate to bind to (p)ppGpp, or YvcI may rapidly hydrolyze (p)ppGpp and release the product. Therefore, we incubated purified YvcI with [5′-α-$^{32}$P]-(p)ppGpp and ran the reaction product using TLC (Fig. S3). We found that YvcI can hydrolyze both ppGpp and pppGpp. We also tested the ability of YvcI to hydrolyze GTP and 8-oxo-GTP to sanitize guanosine nucleotide pool[26]. YvcI failed to hydrolyze either GTP (Fig. S3a) or 8-oxo-GTP (Fig. S3c). The inability of YvcI to hydrolyze GTP despite the structural similarity between GTP and (p)ppGpp suggest that NahA is a specific (p)ppGpp hydrolase which requires its substrate to have pyrophosphate group on the 3′ end.

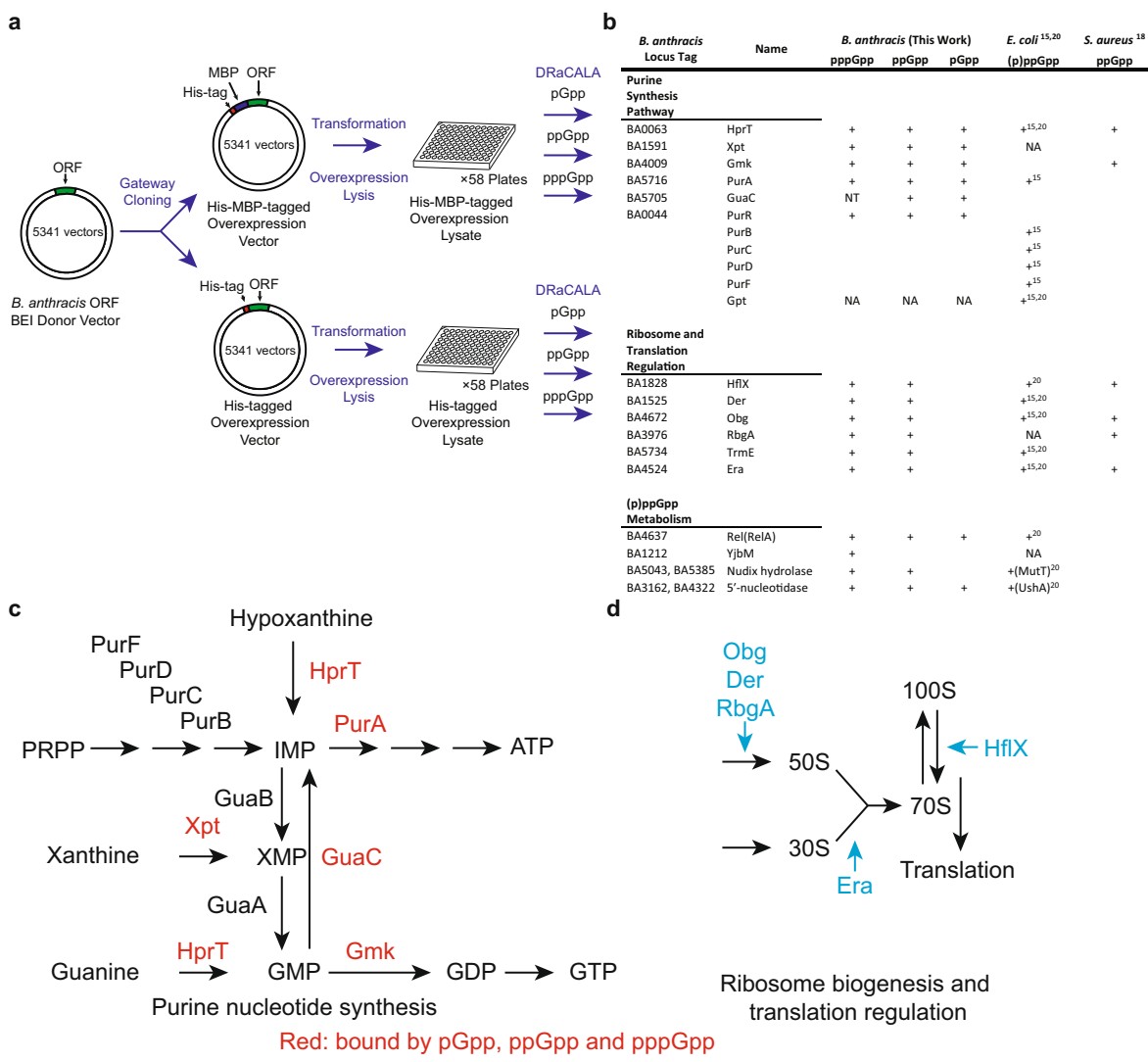

**Fig. 1 Proteome-wide DRaCALA screen identifies both conserved categories of binding targets and novel targets. a** The *Bacillus anthracis* ORF donor vector library was recombined by Gateway cloning into overexpression vectors to generate ORFs with an N-terminal His-tag or HisMBP tag. The plasmids were transformed into *E. coli* for overexpression of recombinant proteins. Lysates of each ORF overexpressed in *E. coli* were assayed for binding to pppGpp, ppGpp, and pGpp using DRaCALA. **b** List of identified (p)ppGpp-binding targets in *E. coli*, *S. aureus*, and *B. anthracis* and pGpp-binding targets in *B. anthracis*. pppGpp results were obtained using both His-tagged and His-MBP-tagged proteins. ppGpp and pGpp results were obtained using His-MBP-tagged proteins. NT not tested, NA not available due to the lack of homologous gene. **c**, **d** Schematics of pathways differentially regulated by pGpp and (p)ppGpp: **c** Enzymes in purine nucleotide synthesis, including HprT, Xpt, Gmk, and GuaC, bound both pGpp and (p)ppGpp; **d** GTPases, involved in ribosome biogenesis and translational control (Obg, HflX, Der, RbgA, and Era), bound (p)ppGpp, but not pGpp.

pppGpp and ppGpp can be hydrolyzed by the Rel enzyme, traditionally referred as RelA in *B. subtilis*[27–29], to produce GTP and GDP respectively. However, unlike Rel, YvcI hydrolyzed pppGpp and ppGpp to yield a single-nucleotide species that migrated differently than GTP (Fig. S3a) or GDP (Fig. S3b). To determine the identity of YvcI's (p)ppGpp hydrolysis product, we analyzed the sample by liquid chromatography coupled with mass spectrometry (LC-MS), and compared to a pGpp standard produced by *E. faecalis* SAS (RelQ) in vitro[22]. The LC-MS profile revealed a peak of the same mass over charge ratio ($m/z$) as GTP but with a different retention time (11.75 min versus 11.15 min for GTP). The retention time is the same as the pGpp standard (Fig. 2b), suggesting that the hydrolysis product of YvcI is pGpp.

The production of pGpp from (p)ppGpp agrees with the NuDiX hydrolase function, inferring that pppGpp and ppGpp are hydrolyzed between the 5′-α and 5′-β phosphate groups to

produce guanosine-5′-monophosphate-3′-diphosphate (pGpp). Therefore, we renamed the enzyme NuDiX alarmone hydrolase A (NahA).

It is possible, although unlikely, that NahA hydrolyzes the 3′-β-phosphate and the 5′-γ-phosphate to produce ppGp, which would run at the same retention time as pGpp in LC-MS. To distinguish these two possibilities, we analyzed the NahA cleavage products of [3′-β-$^{32}$P]-pppGpp. If NahA cleaves between 5′-α and β phosphates, the reaction would yield [3′-β-$^{32}$P]-pGpp. In contrast, if NahA cleaves between 3′-α and β-phosphates, the reaction would yield free $^{32}$P-phosphate. TLC analysis revealed that the radioactive $^{32}$P after NahA hydrolysis of [3′-β-$^{32}$P]-pppGpp co-migrates with the pGpp nucleotide rather than the free phosphate that migrates to the very end of TLC plate (Fig. 2c). This result showed that the product has an intact 3′-pyrophosphate group, confirming the product to be pGpp rather

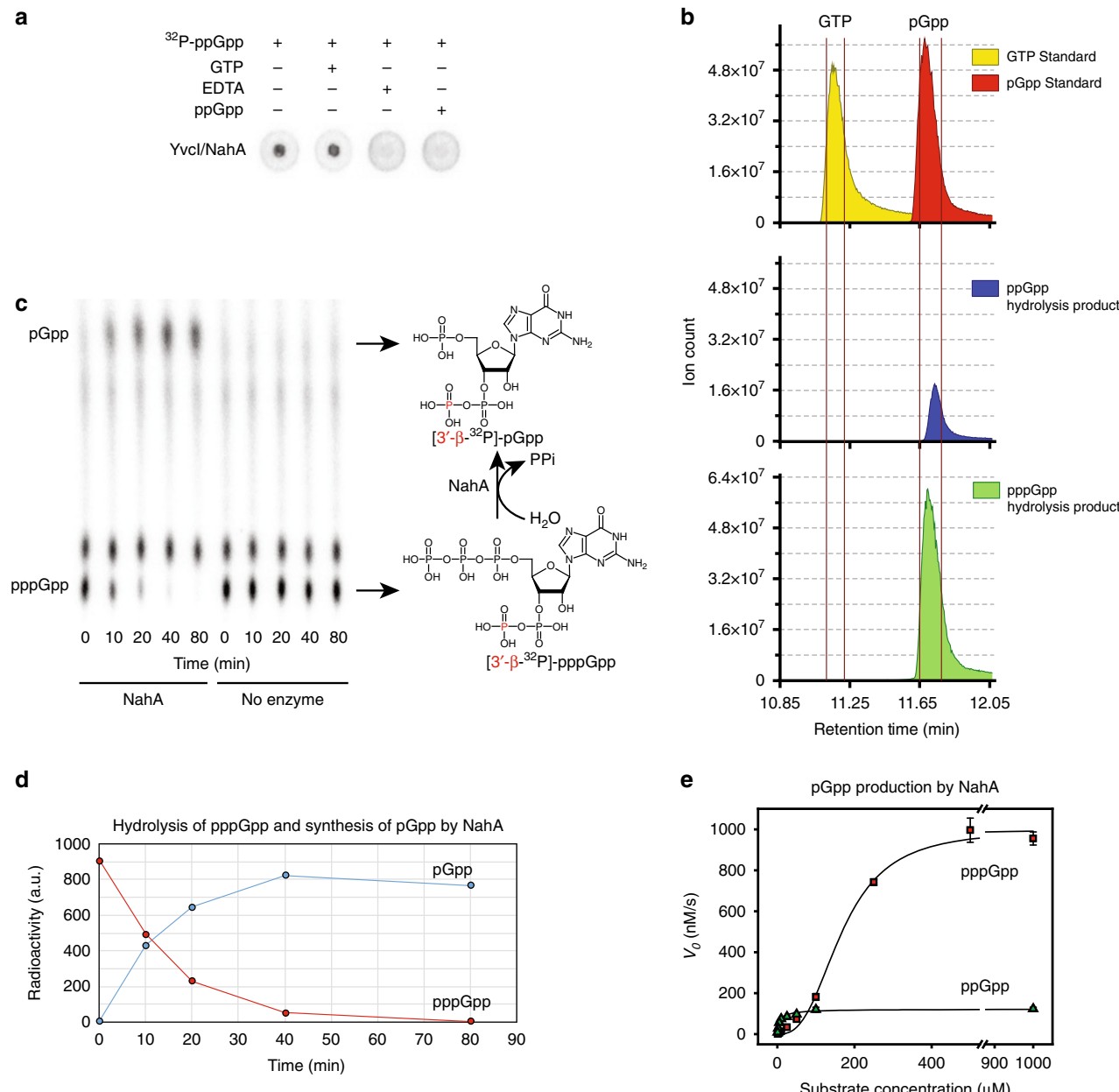

**Fig. 2 NahA (YvcI) produces pGpp via (p)ppGpp hydrolysis. a** DRaCALA of [5′-α-$^{32}$P]-ppGpp binding to *B. subtilis* HisMBP-tagged YvcI (NahA) overexpressed in *E. coli* cell lysate. Unlabeled GTP (100 μM), unlabeled ppGpp (100 μM) and EDTA (10 mM) were added as indicated. **b** Ion count vs. retention time curves from LC-MS of GTP and pGpp standards and NahA-catalyzed hydrolysis products from ppGpp and pppGpp. **c** TLC analysis of NahA activity over time with [3′-β-$^{32}$P]-pppGpp. Expected NahA-catalyzed conversion of [3′-β-$^{32}$P]-pppGpp to [3′-β-$^{32}$P]-pGpp is shown on the right. Radiolabeled [3′-β-$^{32}$P]-phosphorus atom is highlighted in red. **d** Quantitation of pppGpp and pGpp in the hydrolysis of pppGpp in **c**. **e** Initial velocity vs. pppGpp or ppGpp concentration curves for NahA synthesis of pGpp. Data were obtained by kinetic assay using radiolabels (see Methods section). Curves represent the best fit of the data from three independent experiments. Error bars represent standard error of the mean.

than ppGp. Finally, quantification of [3′-β-$^{32}$P]-pppGpp hydrolysis by NahA showed that the decrease of substrate radioactivity mirrored the increase of the single product radioactivity (Fig. 2d), demonstrating the product is exclusively pGpp.

As a NuDiX hydrolase, NahA has been reported to have a modest activity in removing the 5′-phosphate of mRNA[30]. We found that NahA is far more efficient at hydrolyzing (p)ppGpp than at decapping mRNA. Enzymatic assays revealed that NahA hydrolyzes ppGpp following Michaelis–Menten kinetics, with a $k_{cat}$ of 1.22 ± 0.17 s$^{-1}$ and a $K_m$ of 7.5 ± 2.3 μM (Fig. 2e and

**Table 1 Kinetic parameters of (p)ppGpp hydrolysis by NahA.**

|  | ppGpp | pppGpp |
|---|---|---|
| $k_{cat}$ (s$^{-1}$) | 1.22 ± 0.17 | 10.0 ± 0.5 |
| $K_m$ (μM) | 7.5 ± 2.3 | 177.4 ± 0.4 |
| Hill coefficient $n$ | 1 | 2.78 ± 0.47 |

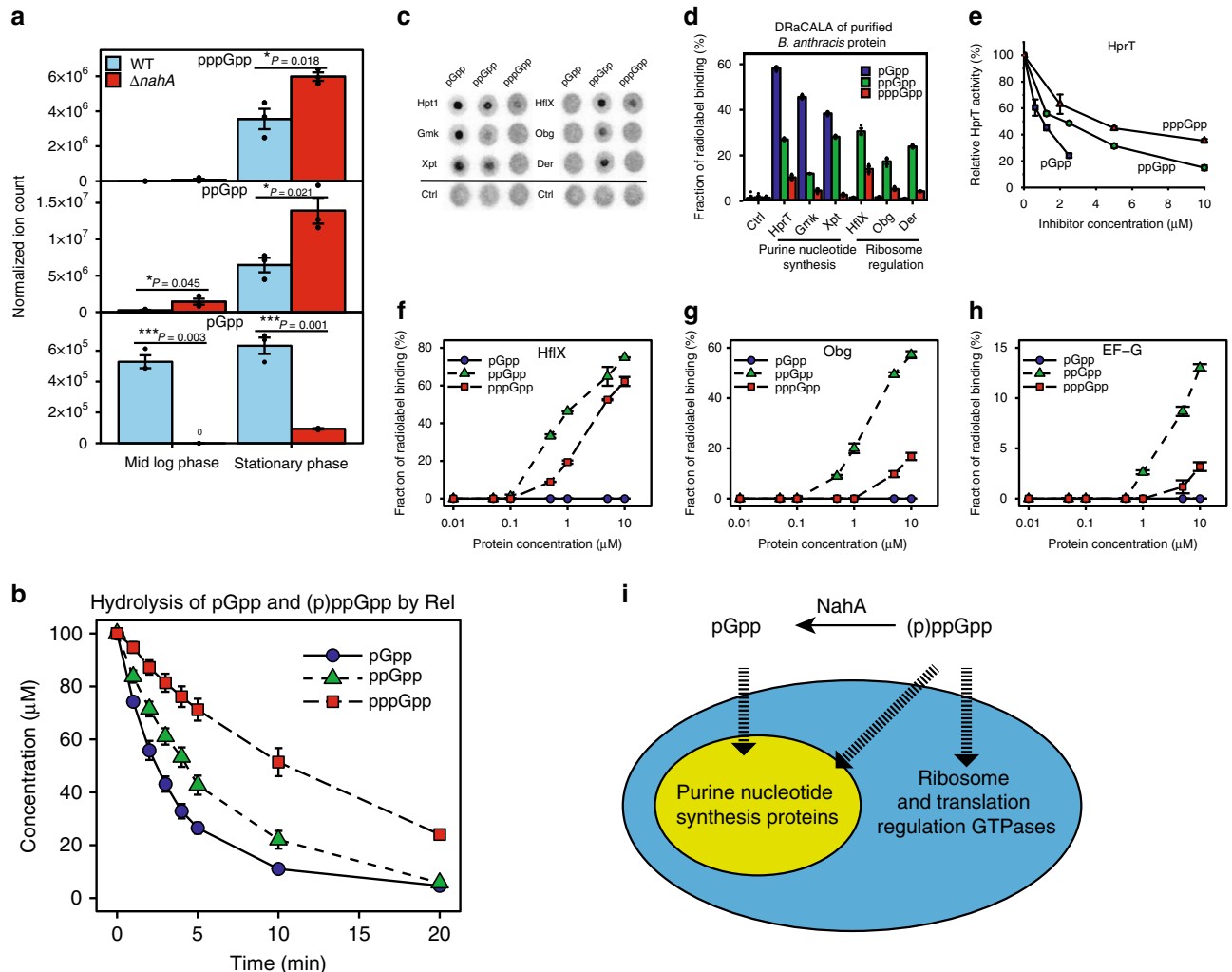

**Fig. 3 pGpp is produced by NahA in vivo and has a distinct binding spectrum compared to (p)ppGpp. a** LC-MS analyses of pGpp, ppGpp, and pppGpp of wild type and $\Delta nahA$ in log phase and stationary phase. Normalized ion count is ion count per $OD_{600nm}$ per unit volume of the culture. Error bars represent standard errors of the mean of three biological replicates. A two-tailed two-sample equal-variance Student's $t$ test was performed between samples indicated by asterisks. Asterisks indicate statistical significance of differences (*$P \leq 0.05$, **$P \leq 0.01$, ***$P \leq 0.001$). **b** In vitro hydrolysis of pGpp, ppGpp, and pppGpp by $B.$ $subtilis$ Rel. For each nucleotide, 200 nM Rel was incubated with a mix of 100 μM unlabeled alarmone and a small amount of 5'-α-$^{32}$P-labeled version. The degradation of the alarmone was analyzed by TLC to measure the decreased radioactivity. Error bars represent standard errors of the mean of three replicates. **c** DRaCALA of purified His-MBP-tagged $B.$ $anthracis$ proteins (1 μM) with <0.1 nM of identical amount of 5'-α-$^{32}$P-labeled pGpp, ppGpp, and pppGpp. **d** Quantification of DRaCALA in **c**. Error bars represent standard errors of the mean of three replicates except for control group (six replicates). **e** HprT enzymatic activities in the presence of indicated concentrations of pGpp, ppGpp, and pppGpp. The reaction was performed with 20 nM HPRT, 1 mM PRPP, and 50 μM guanine[14]. Error bars represent standard errors of the mean from three replicates. **f–h** Titrations of $B.$ $anthracis$ GTPases and quantification of their binding to pGpp, ppGpp and pppGpp using DRaCALA: GTPases HflX (**f**), Obg (**g**), and translation elongation factor G (EF-G) (**h**). Error bars represent standard errors of the mean of three replicates. **i** Schematic showing the relationship between pGpp-binding targets and (p)ppGpp-binding targets.

Table 1). NahA also effectively and cooperatively hydrolyzes pppGpp, with a Hill coefficient of $2.78 \pm 0.47$, $k_{cat}$ of $10.0 \pm 0.5\,\mathrm{s}^{-1}$ and $K_m$ of $177.4 \pm 0.4\,\mu M$ (Fig. 2e and Table 1). In contrast, its $k_{cat}$ to decap RNA is $\sim 0.0003\,\mathrm{s}^{-1}$ (estimation based on published figure)[30]. The difference in $k_{cat}$ in vitro suggests that NahA's major function is to regulate (p)ppGpp rather than to decap the 5' cap of mRNA.

**NahA hydrolyzes (p)ppGpp to produce pGpp in vivo.** NahA was previously identified as a constitutively expressed protein with ~600 copies per cell[31]. To examine its impact on (p)ppGpp in vivo, we engineered a $nahA$ deletion strain, and developed an LC-MS quantification for pppGpp, ppGpp, and pGpp in $B.$ $sub-tilis$ cells (see Methods section). LC-MS measurement of cell extracts showed that $\Delta nahA$ cells accumulate more pppGpp and

ppGpp than wild-type cells during both log phase and stationary phase, in agreement with NahA's ability to hydrolyze (p)ppGpp (Fig. 3a). In contrast, $\Delta nahA$ mutant has much less pGpp than wild-type cells (Fig. 3a). Specifically, during log phase, pGpp can hardly be detected in $\Delta nahA$ (Fig. S4a). When we complement $\Delta nahA$ with an overexpressed copy of $nahA$, the pGpp levels increased to more than wild-type levels (Figure S4). These results support the function of NahA in producing pGpp.

We also used the drug arginine hydroxamate (RHX) which mimics amino acid starvation to induce accumulation of (p)ppGpp[12]. Using both LC-MS and TLC, we observed rapid accumulation of (p)ppGpp after RHX treatment, with $\Delta nahA$ cells showing stronger (p)ppGpp accumulation than wild-type cells (Fig. S5a–d). We still observed the accumulation of pGpp in $\Delta nahA$ cells, although to a much less extent than wild-type cells

(Fig. S5e). This remaining pGpp in Δ*nahA* cells is likely due to the function of the enzyme SAS1, which can synthesize pGpp from GMP and ATP[22].

### pGpp can be hydrolyzed efficiently by *B. subtilis* RelA (Rel) in vitro.

The strong reducing effect of NahA on (p)ppGpp levels suggests that (p)ppGpp are efficiently converted to pGpp. However, pGpp must be eventually degraded. In *B. subtilis*, pppGpp and ppGpp are hydrolyzed to GTP/GDP by RelA (Rel) enzyme, which has a functional (p)ppGpp hydrolase domain. Therefore, we performed in vitro assays using purified *B. subtilis* Rel and pGpp, ppGpp, or pppGpp. We found that all three nucleotides can be hydrolyzed efficiently by Rel, and pGpp was hydrolyzed more rapidly than (p)ppGpp (Fig. 3b).

### Protein binding spectrum of pGpp is distinct from (p)ppGpp.

To understand whether pGpp is just an intermediate of (p)ppGpp hydrolysis or is a bona fide alarmone with its own regulatory targets, we used DRaCALA to systematically screen the *B. anthracis* library for pGpp-binding targets (Fig. 1a, b and Supplementary Data 1). Our screen showed that pGpp binds strongly to multiple purine nucleotide synthesis enzymes (Fig. 1c), but to none of the (p)ppGpp-binding ribosome and translation regulation GTPases (Fig. 1d). We then purified selected pGpp and (p)

ppGpp-binding targets and tested with [5′-α-$^{32}$P]-labeled pGpp, ppGpp, and pppGpp using DRaCALA (Fig. 3c, d). These results confirmed strong pGpp binding to guanosine nucleotide synthesis proteins (Hpt1, Gmk, and Xpt). Enzyme kinetic assay of Xpt[32] and HprT (Fig. 3e) confirmed that pGpp inhibited their activities more potently than ppGpp and pppGpp. In contrast, GTPases involved in ribosome biogenesis (HflX, Obg, Der) bind ppGpp but not pGpp (Fig. 3d). Titration analysis of these GTPases showed their strong affinity to ppGpp, modest affinity to pppGpp and lack of affinity to pGpp (Fig. 3f–h and Supplementary Table 1). We conclude that among the two main groups of conserved interaction targets of (p)ppGpp, pGpp exclusively regulates the purine pathway, but not the GTPases, thus can serve as a specialized signal (Fig. 3i).

### *nahA* mutant exhibits stronger inhibition of translation upon amino acid starvation, delayed outgrowth, and loss of competitive fitness.

The fact that pGpp does not directly regulate ribosome biogenesis and translation implicates an in vivo function of NahA: reducing (p)ppGpp levels to alleviate translation inhibition upon stress, while still keeping purine biosynthesis in check. To test this hypothesis, we analyzed the effects of *nahA* on cellular growth and metabolism. We first compared the key metabolites (NTPs, NDPs, NMPs, nucleosides, and nucleobases)

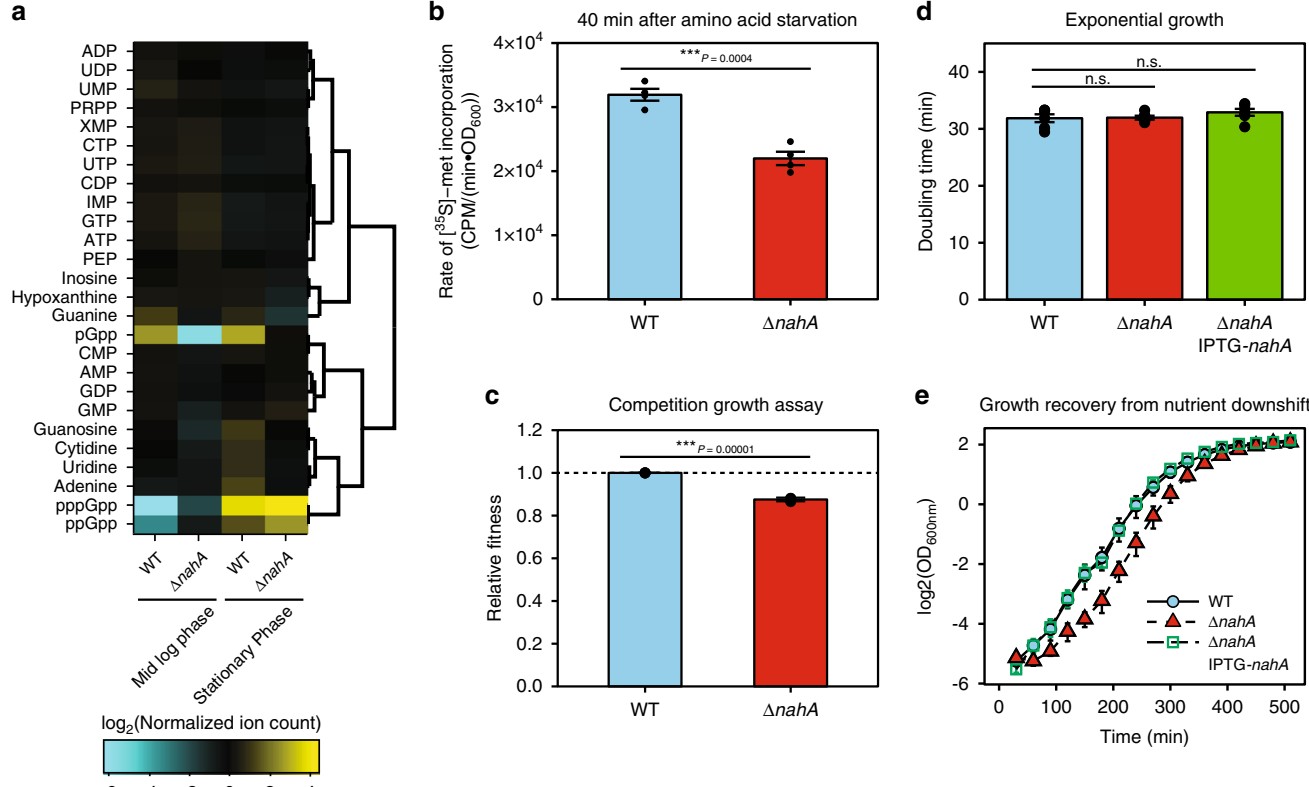

**Fig. 4 The effect of NahA on purine nucleotides, translation, fitness, and growth recovery of *B. subtilis* cells. a** Hierarchical clustering of selected metabolites in exponential growth and stationary phase wild type and Δ*nahA* cells. Metabolites were measured by LC-MS. Normalized ion count is ion count per OD$_{600nm}$ per unit volume of the culture. **b** Protein translation rate of wild type and Δ*nahA* cells 40 min after amino acid starvation. Translation rate was measured by a 5-min pulse of [$^{35}$S]-methionine incorporation into TCA-precipitable fraction. Error bars represent standard errors of the mean from three replicates. **c** Relative fitness of Δ*nahA* and wild type strains obtained from a 7-day serial dilution competition assay. Error bar represents standard deviation from three repeats. **d** Doubling times of Δ*nahA* and wild-type cells in log phase. A two-tailed two-sample equal-variance Student's *t* test was performed between samples indicated by asterisks. Asterisk indicates statistical significance of difference (\**P* ≤ 0.05, \*\*\**P* ≤ 0.001, n.s. *P* > 0.05). **e** Growth recovery from transient nutrient downshift. Log phase cultures of wild type, Δ*nahA* and *nahA* complementation (Δ*nahA* IPTG-*nahA*) strains in rich media with 20 amino acids were firstly downshifted in minimum media without amino acids for 10 min, and then diluted in rich media for outgrowth. Result of growth recovery from similar downshift with the presence of guanosine is shown in Figure S6c. Error bars in **d**, **e** represent standard errors from six replicates.

between wild-type cells and the *nahA* mutant, using LC-MS analyses of exponential growth and stationary phase cells (Fig. 4a and Supplementary Data 2). Despite much higher (p)ppGpp levels and much lower pGpp levels in the *nahA* mutant, most purine nucleotides exhibit very little difference, corroborating with the observation that pGpp regulates purine metabolism similarly to (p)ppGpp.

Next, we examined the effect of *nahA* on protein translation. We measured total protein translation rate of wild type and ΔnahA *B. subtilis* by pulsed incorporation of $^{35}$S-methionine (Fig. 4b). We induced amino acid starvation either by resuspending cells from amino acid replete medium to minimal medium, or by adding the nonfunctional amino acid analog arginine hydroxamate. In both cases, the rate of $^{35}$S-methionine incorporation is significantly lower in ΔnahA than in wild type, indicating a stronger inhibition of translation (Figs. 4b and S6a).

Finally, we examined the impact of *nahA* on fitness of *B. subtilis*. We performed a growth competition assay in which a mixture of ΔnahA and wild-type cells were grown to saturation and then diluted repeatedly for days. The proportion of ΔnahA rapidly decreased (Fig. S6b). The relative fitness of ΔnahA is 0.876 ± 0.009 (mean ± S.D.), indicating significant loss of fitness compared to wild-type cells (Fig. 4c). We found that ΔnahA has a similar doubling time as wild-type cells (Fig. 4d), but a longer lag phase in adjusting to starvation (Fig. 4e). Together, these results suggest that NahA tunes alarmone composition and alarmone levels to promote *B. subtilis* adjustment to nutrient fluctuation and optimizes growth and fitness.

## Discussion

Alarmones are universal stress signaling nucleotides in bacteria, however, the repertoire of alarmones and the target spectrums for each alarmone are incompletely understood. In this work, we have comprehensively characterized the interactomes of the related alarmones pppGpp and ppGpp and established the in vivo presence of pGpp as a closely related alarmone in Gram-positive *Bacillus* species. We characterized the direct targets of (p)ppGpp by screening an open reading frame expression library from *B. anthracis*. From this screen, we identified an enzyme NahA that converts (p)ppGpp to pGpp as efficient means to produce pGpp and to reduce (p)ppGpp concentrations, thus regulating the composition of the alarmones. We demonstrated that pGpp binds a distinct subset of protein receptors of (p)ppGpp. We also identified a key role of NahA in nutrient adaptation, suggesting that regulating alarmone composition may serve as a separation-of-function strategy for optimal adaptation.

**Conservation of pppGpp and ppGpp regulation of purine biosynthesis and ribosome biogenesis pathways across different species of bacteria**. (p)ppGpp regulates diverse cellular targets that differ between different bacteria. For example, (p)ppGpp directly binds to RNA polymerase in *E. coli* to control the transcription of ribosomal and tRNA operons yet the (p)ppGpp-binding sites on RNA polymerase are not conserved beyond proteobacteria[4–8]. Instead, (p)ppGpp accumulation in firmicutes strongly down-regulates synthesis of GTP, the exclusive transcription initiating nucleotides of rRNA and tRNA operons in firmicutes, to achieve a similar transcription control with different direct targets[12,33,34]. Therefore, identifying whether certain aspects of (p)ppGpp regulation are conserved among bacterial species is important for understanding the principles of bacterial survival and adaptation.

Our DRaCALA screen with a *B. anthracis* library revealed many ppGpp and pppGpp binding targets in this pathogenic Gram-positive bacterium. Comparing ppGpp-binding targets in

*B. anthracis* to *S. aureus*[18] and *E. coli*[15,20] identified novel targets but more importantly, revealed a clear theme of conservation. Most notably, (p)ppGpp in all three species binds to multiple proteins in two key pathways: (1) purine nucleotide synthesis, (2) translation-related GTPases including ribosome biogenesis factors.

The regulation of purine nucleotide synthesis in firmicutes includes well characterized targets Gmk and HprT whose regulation by (p)ppGpp protects *Bacillus subtilis* against nutrient changes like amino acid starvation and purine fluctuation, preventing cells from accumulating toxic high levels of intracellular GTP[12,14,35]. These also include new-found likely targets such as enzymes GuaC and PurA and the de novo pathway transcription factor PurR. Intriguingly, in the evolutionarily distant *E. coli*, the ppGpp targets in the purine biosynthesis pathway are different. For example, Gmk is not regulated by (p)ppGpp in *E. coli*[35]. On the other hand, (p)ppGpp directly targets the *E. coli* de novo enzyme PurF[15], a target that is not conserved in firmicutes. Therefore, despite differences in precise targets, (p)ppGpp extensively regulates the purine biosynthesis pathway in evolutionarily diverse bacteria (Fig. 1c), highlighting this critical physiological role of (p)ppGpp.

We also found that (p)ppGpp interacts with essential GTPases that are implicated in ribosome biogenesis and translation in *B. subtilis* and *B. anthracis* (Fig. 1d). (p)ppGpp's targets in GTPases from *E. coli* to firmicutes are conserved[16–19]: HflX, Obg, and Era, are also (p)ppGpp-binding proteins in *E. coli* and *S. aureus*[15,18,20]; Der and TrmE were identified in *E. coli* although not in *S. aureus* as (p)ppGpp-binding proteins[15,20]; RbgA does not exist in *E. coli* and it is identified as a (p)ppGpp-binding protein in *S. aureus*[18]. HflX is a ribosome-splitting factor which may rescue stalled ribosomes under stressed conditions[36], and mediates the dissociation of hibernating 100S ribosome to resume normal translation[37]. Obg[16], Der[38], and RbgA[39,40] participate in the maturation of the 50 S subunit of ribosome. Era functions in the assembly of the functional 70 S ribosome complex[41]. TrmE functions in the maturation of tRNA to facilitate translation[42]. (p)ppGpp can be produced by amino acid starvation-induced translational stress or by defects in tRNA maturation[43]. Thus the conservation of (p)ppGpp regulation of GTPase targets highlights the key function of (p)ppGpp in quality control of ribosome biogenesis and regulation of translation.

**The third major alarmone pGpp in Gram-positive species and its specific regulatory effect**. In addition to pppGpp and ppGpp, it was long suspected that in *B. subtilis*, there are additional alarmones such as pppApp, pGpp, and ppGp accumulating during the stringent response[44,45]. Here we detected pGpp in *B. subtilis* cells, characterized an enzyme for its production in vivo and in vitro, and identified its targets. In contrast to our DRaCALA screens for pppGpp and ppGpp which identify similar targets and functionalities between these two alarmones, our DRaCALA screen for pGpp displays a strong difference from (p)ppGpp with regarding to two conserved pathways. The affinity and inhibitory potency of pGpp for purine nucleotide synthesis enzymes are equivalent or higher than that of (p)ppGpp. In contrast, pGpp's affinity to GTPases involved in translational regulation is much lower, or completely absent, compared to (p)ppGpp. The distinct profile of target receptors establishes pGpp as a different alarmone from (p)ppGpp, allowing fine tuning of bacterial stress response. We propose a model for the function of NahA in growth recovery and competitive fitness by its role in transforming the alarmones. In wild-type cells, (p)ppGpp produced in response to amino acid starvation will be hydrolyzed in part by NahA to pGpp. (p)ppGpp concentration in wild-type cells

during amino acid starvation can reach >1 mM. Given the copy number of NahA as ~600 copies per cell and its maximum velocities for (p)ppGpp hydrolysis, (p)ppGpp concentrations observed in wild-type cells is a balance of synthesis by enzymes Rel and SAS1/2, and hydrolysis by Rel and NahA. Thus, in Δ*nahA* cells, (p)ppGpp accumulates higher than in wild-type cells during amino acid starvation. Stronger inhibition of translation due to overly high level of (p)ppGpp in Δ*nahA* leads to slower growth recovery after nutrient stress and thus leads to fitness loss when co-cultured with wild-type strain. In addition, because pGpp appears to be hydrolyzed more efficiently by RelA than ppGpp and pppGpp (Fig. 3b), another role of NahA is perhaps to speed up (p)ppGpp removal to promote growth recovery.

In *E. coli*, (p)ppGpp-binding proteins also include NuDiX hydrolases NudG and MutT[20]. Like NahA, NudG and MutT also hydrolyze (p)ppGpp. Unlike NahA, NudG and MutT produce guanosine 5′-monophosphate 3′-monophosphate (pGp) rather than pGpp. Similarly, in the bacterium *Thermus thermophilus*, the NuDiX hydrolase Ndx8 is also found to hydrolyze (p)ppGpp to produce pGp[46]. Sequence alignment shows that their homology with NahA is mostly restricted in the NuDiX box that is shared by NuDiX hydrolase family (Fig. S1). Therefore, NudG, MutT, and Ndx8 are considered alternative (p)ppGpp hydrolysis pathways to remove (p)ppGpp and promote growth[46], rather than an alarmone-producer.

In *E. coli*, the enzyme GppA converts pppGpp to ppGpp, which may also regulate the composition of its alarmones[21]. In the absence of GppA, the alarmone pppGpp accumulates to a higher level than ppGpp[47]. It is possible that a similar separation-of-function regulation exists in *E. coli* by tuning pppGpp vs ppGpp levels. This can be addressed by examining pppGpp interactome in *E. coli*, which may reveal a differential theme than ppGpp. Ultimately, discovery of more enzymes that interconvert signaling molecules may reveal a universal theme of optimization through fine-tuning in signal transduction.

## Methods

**Bacillus anthracis ORFeome Library Construction.** *Bacillus anthracis* Gateway® Clone Set containing plasmids bearing *B. anthracis* open reading frames was acquired from BEI Resources and used for Gateway cloning (Invitrogen protocol) into overexpression vectors pVL791[48] (10xHis tag ampicillin-resistant) and pVL847[48] (10xHis-MBP tag, gentamicin-resistant) and transformed into *Escherichia coli* BL21 lacI$^q$ to produce two open reading frame proteome over-expression libraries (ORFeome library). The resulting ORFeome library contains 5139 ORFs from the genome of *B. anthracis* str. *Ames* (91.2% of 5632 ORFs in the genome). The corresponding proteins were expressed in *E. coli* and the cells were lysed to prepare the overexpression lysates for the downstream analysis: *E. coli* strains containing overexpression vectors were grown in LB-M9 media (7 g/L Na$_2$HPO$_4$, 2 g/L KH$_2$PO$_4$, 0.5 g/L NaCl, 1 g/L NH$_4$Cl, 2 g/L glucose, 1 g/L sodium succinate dibasic hexahydrate; 10 g/L tryptone, 5 g/L yeast extract, and 3 mM MgSO$_4$) at 30 °C and the overexpression was induced by 0.5 mM IPTG. Cells were collected by centrifugation at 5000×*g* 4 °C for 10 min. Cells were resuspended in 10% original volume of resuspension/lysis buffer (10 mM Tris pH 7.5, 100 mM NaCl, 5 mM MgCl$_2$, 0.25 mg/mL lysozyme, 0.01 mg/mL DNase I, and 1 mM PMSF). Cells were then lysed by three rounds of −80 °C freezing and room temperature thawing.

**Plasmid and strain construction.** Plasmid for NahA purification was constructed as follows. To generate a His-SUMO-NahA plasmid, the *nahA* sequence was PCR amplified using primers oJW3519/oJW3520 and incorporated into pE-SUMO vector backbone amplified using primers oJW3194/oJW3195 by Golden Gate assembly method (New England BioLabs) to generate pJW739.

To generate a His-MBP-NahA plasmid, the *nahA* sequence was PCR amplified using primers oJW3274/oJW3275 and incorporated into pVL847[48] (His-MBP-tag overexpression vector) by Gateway cloning method (Invitrogen) to generate pJW742.

Δ*nahA*::*erm* mutant of *Bacillus subtilis* was constructed by transformation of PCR product of BKE34780 (from BGSC) genome DNA amplified by oJW3382/oJW3383 and erythromycin-lincomycin selection.

Δ*nahA* markerless deletion of *Bacillus subtilis* was constructed by CRISPR/Cas9 editing method[49]: CRISPR editing plasmid pJW737 was constructed by Golden Gate assembly method (New England BioLabs) using pPB41 backbone, guide RNA template, and DNA fragments downstream and upstream of *nahA*. pPB41[49]

### Table 2 Strains used in this work.

| ID | Genotype | Ref. |
|---|---|---|
| DK1042 | *Bacillus subtilis* NCIB 3610 comI$^{Q12L}$ (wild type) | 55 |
| JDW4085 | *Bacillus subtilis* NCIB 3610 comI$^{Q12L}$ Δ*nahA*::*erm* | This work |
| JDW4087 | *Bacillus subtilis* NCIB 3610 comI$^{Q12L}$ Δ*nahA* | This work |
| JDW4088 | *Bacillus subtilis* NCIB 3610 comI$^{Q12L}$ Δ*nahA* amyE::Phyperspank::nahA | This work |

### Table 3 Plasmids used in this work.

| ID | Construct | Source |
|---|---|---|
| pJW736 | pDR111 *amyE::Phyperspank::nahA* (amp, spec) | This work |
| pJW737 | pPB41-gRNA(*nahA*)-Δ*nahA* (amp, spec) | This work |
| pJW739 | pE-SUMO-*Bacillus subtilis nahA* (kan) | This work |
| pJW742 | pVL847-*Bacillus subtilis nahA* (gentamicin) | This work |

backbone was amplified by oJW2775/oJW2821. Guide RNA template was made by annealing oJW3501/oJW3502. *B. subtilis* DNA fragments upstream and downstream of *nahA* were amplified by oJW3498/oJW3500 and oJW3497/oJW3499, respectively. pJW737 was transformed into wild type *B. subtilis* followed 30 °C spectinomycin selection. Transformants were sequentially patched and grown on LB agar without antibiotics at 45 °C to remove the plasmid pJW737. Colonies cured of pJW737 (sensitive to spectinomycin) were PCR verified using oJW3382/oJW3383 and the colonies with PCR product of 1422 bp were Δ*nahA*.

To generate an IPTG-inducible *nahA* in *B. subtilis*, the *nahA* sequence was PCR amplified using primers oJW3400/oJW3401 and incorporated into pDR111[50] by restriction cut by SalI and SphI followed by T4 DNA ligase mediated ligation (New England BioLabs) to generate pJW736. Δ*nahA* amyE::Phyperspank::nahA (Δ*nahA* IPTG-*nahA*) mutant of *Bacillus subtilis* was constructed by transformation of pJW736 into Δ*nahA* mutant and spectinomycin selection.

Strains, plasmids, and primers are listed in (Tables 2–4).

**Growth conditions.** If not specifically mentioned, *B. subtilis* strains were grown in S7 defined media[51] with modifications (50 mM MOPS instead of 100 mM, 0.1% potassium glutamate, 1% glucose, no additional amino acids, and nucleosides or nucleobases), with shaking at 250 rpm 37 °C. The modified S7 defined media with 20 amino acids contains 50 μg/mL alanine, 50 μg/mL arginine, 50 μg/mL asparagine, 50 μg/mL glutamine, 50 μg/mL histidine, 50 μg/mL lysine, 50 μg/mL proline, 50 μg/mL serine, 50 μg/mL threonine, 50 μg/mL glycine, 50 μg/mL isoleucine, 50 μg/mL leucine, 50 μg/mL methionine, 50 μg/mL valine, 50 μg/mL phenylalanine, 500 μg/mL aspartic acid, 500 μg/mL glutamic acid, 20 μg/mL tryptophan, 20 μg/mL tyrosine, and 40 μg/mL cysteine.

**Nucleotide preparation.** pppGpp was synthesized in vitro from 8 mM ATP and 6 mM GTP using Rel*Seq*$_{1-385}$[21]. For production of [5′-α-$^{32}$P]-pppGpp, 750 μCi/mL $^{32}$P-α-GTP (3000 mCi/mmol; PerkinElmer) was used instead of non-radiolabeled GTP. For production of [3′-β-$^{32}$P]-pppGpp, 750 μCi/mL $^{32}$P-γ-ATP (3000 mCi/mmol; PerkinElmer) was used instead of non-radiolabeled ATP. ppGpp was synthesized from pppGpp using GppA[21]. For purification of nucleotides, the reaction mix was diluted in Buffer A (0.1 mM LiCl, 0.5 mM EDTA, 25 mM Tris pH 7.5). The mixture was loaded onto a Buffer-A-equilibrated 1 mL HiTrap QFF column (GE Healthcare). The column was washed by 10 column volumes of Buffer A, followed by 10 column volumes of Buffer A with 170 mM LiCl for pppGpp purification (for ppGpp purification, the LiCl concentration was 160 mM). Radiolabeled (p)ppGpp was eluted by 5 column volumes of Buffer A with 500 mM LiCl with 1 mL fractions.

pGpp was synthesized from pppGpp in vitro using NahA in a reaction containing 25 mM bis-Tris propane pH 9.0, 15 mM MgCl$_2$, 75 mM NH$_4$Cl and 0.1 mg/mL NahA. For purification, we diluted the reaction mix in Buffer A. The mixture was loaded onto a Buffer-A-equilibrated 1 mL HiTrap QFF column (GE Healthcare), washed by 10 column volumes of Buffer A, and eluted by 10 column volumes of Buffer A with 155 mM LiCl with 1 mL fractions, followed by 5 column volumes of Buffer A with 500 mM LiCl with 1 mL fractions. The radioactivity and purity of radiolabeled nucleotides were analyzed by thin-layer chromatography and phosphorimaging.

**Overexpression and purification of NahA.** His-tagged NahA was purified by Ni-NTA affinity column, followed by SUMO protease cleavage to remove the tag. To express His-tagged protein, His-SUMO-NahA verctor (pJW739) was transformed into *E. coli* BL21(DE3) lacI$^q$ by chemical transformation. A single colony of the corresponding strain was grown in LB with 30 μg/mL kanamycin overnight.

**Table 4 Primers used in this work.**

| ID | Sequence |
|---|---|
| oJW2775 | GACGGTCTCAGCTGGCTGTAGGCATAGGCTTGGTTATG |
| oJW2821 | GTAGGTCTCTAAGGATTTCGCGGGATCGAGATCCTGCATTAATG |
| oJW3194 | GCGGGTCTCAACCTCCAATCTGTTCGCGGTGAGCC |
| oJW3195 | GCGGGTCTCATAATCGAGCACCACCACCACCACCA |
| oJW3274 | GGGGACAAGTTTGTACAAAAAAGCAGGCTGGGTGACGTACTTGCAAAGAGTGACAAATTG |
| oJW3275 | GGGGACCACTTTGTACAAGAAAGCTGGGTCTATTTGATGTGCTGCGGGTCTAAACGAT |
| oJW3382 | GCTCAAAGTATTCTTCAAGCGAGAG |
| oJW3383 | CATTCCACTTCATGACGTAAGAGG |
| oJW3400 | CCCGTCGACAAAGGAGGTGTACATGTGACGTACTTGCAAAGAGTGACAAATTGT |
| oJW3401 | CCCGCATGCCTATTTGATGTGCTGCGGGTCTAAACGATA |
| oJW3497 | CAGGGTCTCACCTTCAAGCGGCAGGCCAGTCGCTGCCAGCGGAT |
| oJW3498 | GTCGGTCTCACAGCCCAAATCGTAACGGCTACAGGAGACGGAAG |
| oJW3499 | GTCGGTCTCTTTAGAAAGACAAGTCAGGGGGGAGAAAGA |
| oJW3500 | GACGGTCTCACTAACCTTCGTCCTGTCATCGTCTCTTTAT |
| oJW3501 | GACGGTCTCAAAACATACCAGTCTCTTCTCTGTACTCTCTGATGAGTTTTAGAGACCGTC |
| oJW3502 | GACGGTCTCTAAAACTCATCAGAGAGTACAGAGAAGAGACTGGTATGTTTTGAGACCGTC |
| oJW3519 | GTGGGTCTCTAGGTGTGACGTACTTGCAAAGAGTGACAAATTGT |
| oJW3520 | GTGGGTCTCTATTATTTGATGTGCTGCGGGTCTAAACGATA |

Overnight culture was 1:100 diluted in LB-M9 media with 30 μg/mL kanamycin and grown at 30 °C with shaking for 4 h. After adding 1 mM IPTG for induction, the culture was further grown at 30 °C for 4 h. Then the culture was centrifuged at 4000×g for 30 min at 4 °C and the supernatant was discarded. The pellet was stored at −80 °C before cell lysis. All the following steps were performed at 4 °C. Pellet was resuspended in Lysis Buffer (50 mM Tris-HCl pH 8, 10% sucrose w/v, and 300 mM NaCl) and lysed by French press. Cell lysate was centrifuged at 15,000 rpm for 30 min, and the supernatant was collected, filtered through 0.45-μm pore-size cellulose syringe filter. His-SUMO-NahA filtered supernatant was purified using a 5-mL HisTrap FF column (GE Healthcare) equipped on an ÄKTA FPLC apparatus (GE Healthcare). SUMO Buffer A (50 mM Tris-HCl, 25 mM imidazole, 500 mM NaCl, and 5% glycerol v/v) was used as washing buffer, and SUMO Buffer C (50 mM Tris-HCl, 500 mM imidazole, 500 mM NaCl, and 5% glycerol v/v) was used as the elution buffer. Fractions containing most abundant His-SUMO-NahA were combined with 100 μg SUMO Protease and dialyzed twice against SUMO Protease Buffer (50 mM Tris-HCl, 500 mM NaCl, 5% glycerol v/v, and 1 mM β-mercaptoethanol) overnight. His-SUMO tag was removed by flowing through HisTrap FF column and collecting the flow-through. Purified NahA was analyzed by SDS-PAGE and the concentration was measured by the Bradford assay (Bio-rad).

His-MBP-NahA was prepared using similar protocol as mentioned above with modifications: pJW742 was transformed into *E. coli* and all the media contain 10 μg/mL gentamycin instead of kanamycin; SUMO-protease and His-SUMO tag removal were not applied.

**Differential radial capillary action ligand assay**. Cell lysate with overexpressed protein and purified protein were used for DRaCALA. In all, 10 μL cell lysate or diluted, purified protein was mixed with 10 μL diluted [5′-α-$^{32}$P]-(p)ppGpp or [5′-α-$^{32}$P]-pGpp (~0.2 nM) in a buffer containing 10 mM Tris pH 7.5, 100 mM NaCl and 5 mM MgCl$_2$, incubated at room temperature for 10 min. In total, ~2 μL mixture was blotted onto nitrocellulose membrane (Amersham; GE Healthcare) and allowed for diffusion and drying. The nitrocellulose membrane loaded with mixture was exposed on phosphor screen, which was scanned by a Typhoon FLA9000 scanner (GE Healthcare). Fraction of (p)ppGpp binding $F$ was analyzed using the following equation[24]:

$$F = \frac{I_{in} - I_{bkg}}{I_{tot}} \quad (1)$$

where $I_{in}$ is the intensity of the inner region of radioactivity pattern, $I_{bkg}$ is the background intensity in the inner region representing the unbound ligands, and $I_{tot}$ is the total intensity of the whole radioactivity pattern. $I_{bkg}$ can be calculated by the following equation:

$$I_{bkg} = A_{in} \times \frac{I_{tot} - I_{in}}{A_{tot} - A_{in}} \quad (2)$$

where $A_{in}$ is the area of the inner region of radioactivity pattern, and $A_{tot}$ is the total area of the whole radioactivity pattern.

**Quantification of intracellular nucleotides by thin-layer chromatography**. Cells were grown in low phosphate S7 defined media (0.5 mM phosphate instead of 5 mM), labeled with 50 μCi/mL culture $^{32}$P-orthophosphate at an optical density at 600 nm (OD$_{600nm}$) of ~0.01 and grown for two to three additional generations before sampling. Nucleotides were extracted by adding 100 μL of sample into 20 μL 2 N formic acid on ice for at least 20 min. Extracted samples were centrifuged at 15,000×g for 30 min to remove cell debris. Supernatant samples were spotted on

PEI cellulose plates (EMD-Millipore) and developed in 1.5 M or 0.85 M potassium phosphate monobasic (KH$_2$PO$_4$) (pH 3.4). TLC plates were exposed on phosphor screens, which were scanned by a Typhoon FLA9000 scanner (GE Healthcare). Nucleotide levels were quantified by ImageJ (NIH). Nucleotide levels are normalized to ATP level at time zero.

**Kinetic assay of pppGpp, ppGpp, and pGpp hydrolysis**. Hydrolysis reaction was performed at 37 °C. For NahA hydrolysis of (p)ppGpp, 100 nM purified NahA was added to a reaction mix containing 40 mM Tris-HCl pH 7.5, 100 mM NaCl, 10 mM MgCl$_2$, indicated concentrations of non-radioactive and $^{32}$P-radiolabeled (p) ppGpp. For (p)ppGpp and pGpp hydrolysis by *B. subtilis* RelA, 200 nM purified RelA enzyme was added to a reaction mix containing 25 mM Tris-HCl pH 7.5, 1 mM MnCl$_2$, 100 μM non-radioactive, and ~0.2 nM $^{32}$P-radiolabeled pppGpp, ppGpp, or pGpp.

At each indicated time points after the reaction started, 10 μL reaction mix was aliquoted into 10 μL ice-chilled 0.8 M formic acid. Samples at each time point were resolved by thin-layer chromatography on PEI-cellulose plates with 1.5 M KH$_2$PO$_4$ (pH 3.4). Nucleotide levels were quantified as mentioned above and the phosphorimager counts of substrate and product were used to calculate the concentration of product by a formula:

$$c_P = c_{S0} \cdot \frac{V_P}{V_P + V_S} \quad (3)$$

Here $c_P$ was the concentration of product, $c_{S0}$ was the concentration of substrate before the reaction starts, $V_P$ was the phosphorimager count of product, and $V_S$ was the phosphorimager count of substrate. Initial rates of hydrolysis ($v_0$) were calculated using the slope of the initial linear part of $c_P$ over time curve at different initial substrate concentrations. Michaelis–Menten constant ($K_m$) and catalytic rate constant ($k_{cat}$) were obtained by fitting the data of $v_0$-$c_{S0}$ to the model

$$v_0 = \frac{c_E \cdot k_{cat} \cdot c_{S0}^n}{K_m^n + c_{S0}^n} \quad (4)$$

by MATLAB (R2016b), where $c_E$ was the concentration of NahA, $c_{S0}$ was the initial concentration of substrate, and $n$ was the Hill's coefficient (for ppGpp hydrolysis, fix $n$ to 1).

**LC-MS quantification of metabolites**. Cells were grown to designated OD$_{600nm}$. In all, 5 mL of cultures were sampled and filtered through PTFE membrane (Sartorius) at time points before and after 0.5 mg/mL arginine hydroxamate treatment. Membranes with cell pellet were submerged in 3 mL extraction solvent mix (on ice 50:50 (v/v) chloroform/water) to quench metabolism, lyse the cells and extract metabolites. Mixture of cell extracts were centrifuged at 5000×g for 10 min to remove organic phase, then centrifuged at 20,000×g for 10 min to remove cell debris. Samples were frozen at −80 °C if not analyzed immediately. Samples were analyzed using an HPLC-tandem MS (HPLC-MS/MS) system consisting of a Vanquish UHPLC system linked to heated electrospray ionization (ESI, negative mode) to a hybrid quadrupole high resolution mass spectrometer (Q-Exactive orbitrap, Thermo Scientific) operated in full-scan selected ion monitoring (MS-SIM) mode to detect targeted metabolites based on their accurate masses. MS parameters were set to a resolution of 70,000, an automatic gain control (AGC) of 1e6, a maximum injection time of 40 ms, and a scan range of 90–1000 mz. LC was performed on an Aquity UPLC BEH C18 column (1.7 μm, 2.1 × 100 mm; Waters). Total run time was 30 min with a flow rate of 0.2 mL/min, using Solvent A (97:3 (v/ v) water/methanol, 10 mM tributylamine pH~8.2–8.5 adjusted with ~9 mM acetic

acid) and 100% acetonitrile as Solvent B. The gradient was as follows: 0 min, 5% B; 2.5 min, 5% B; 19 min, 100% B; 23.5 min 100% B; 24 min, 5% B; and 30 min, 5% B. Raw output data from the MS was converted to mzXML format using inhouse-developed software, and quantification of metabolites were performed by using the Metabolomics Analysis and Visualization Engine (MAVEN 2011.6.17, http://genomics-pubs.princeton.edu/mzroll/index.php) software suite[52,53]. Normalized ion count was defined and calculated as the ion count per $OD_{600nm}$ per unit volume (5 mL) of the culture.

**Growth competition assay.** $nahA::erm^R$ mutant and wild-type cells were mixed in LB broth to an $OD_{600nm}$ of 0.03 and grown at 37 °C with vigorous shaking. After every 24-h period, the stationary phase culture was back-diluted in fresh LB broth to an $OD_{600nm}$ of 0.02, for a total period of 7 days. Each day, the culture was sampled, serially diluted, and spread over LB agar and LB agar containing 0.5 μg/mL erythromycin/12.5 μg/mL lincomycin to obtain the CFU of total bacteria and erythromycin-resistant strain, respectively. Relative fitness was calculated by the formula[54]:

$$w = \frac{\log_2\left(CFU_{erm_t^R} / CFU_{erm_0^R}\right)}{\log_2\left(CFU_{erm_t^S} / CFU_{erm_0^S}\right)} \quad (5)$$

where $w$ means relative fitness of the erythromycin-resistant strain to the erythromycin-sensitive strain; $CFU_{erm_0^R}$ and $CFU_{erm_t^R}$ means the total CFU of erythromycin-resistant strain before and after competition, respectively; $CFU_{erm_0^S}$ and $CFU_{erm_t^S}$ means the total CFU of erythromycin-sensitive strain before and after competition, respectively. The CFUs were adjusted according to the dilution factors of the back-dilutions.

**[35S]-Methionine incorporation assay.** Wild type and $nahA$ mutant cells were grown in defined S7 medium with glucose and 20 amino acids (concentrations listed in the "Plasmid and Strain Construction and Growth Conditions" section) to $OD_{600nm}$ ~0.3. For amino acid starvation, cells were pelleted and resuspended in S7 glucose medium without amino acid. Alternatively, cells were treated with 0.5 mg/ml arginine hydroxamate. Aliquots of 200 μL culture were taken at indicated time points after treatment to label with 0.025 μCi/μL [35S]-Methionine for 5 min, before adding 200 μL ice-chilled 20% (w/v) trichloroacetic acid. Samples were chilled before filtration. Glass fiber filters (24 mm, GF6, Whatman) were prewetted with 3 mL ice-chilled 5%(w/v) trichloroacetic acid and applied with samples. Filters were then washed three times by 3 × 10 mL ice-chilled 5%(w/v) trichloroacetic acid and dried by ethanol. Dried filters were put in scintillation vials, mixed with 5 mL scintillation fluid and then sent for scintillation count in the range of 2.0-18.6 eV. Counts per minutes (CPM) measured by scintillation counter, divided by the period of labeling and the $OD_{600nm}$ of the corresponding culture, were used as the representative of translation rate.

**Growth recovery from nutrient downshift.** Cells were grown in defined amino acid repleted medium (S7 medium with glucose, 20 amino acids, and 0.5 mM IPTG) to mid log phase ($OD_{600 nm}$~0.5), and then washed in S7 glucose without amino acid. Washed cultures were diluted into fresh medium with 20 amino acids and 0.5 mM IPTG to $OD_{600 nm} = 0.01$ and the growth was monitored by a plate reader (Synergy 2, Biotek) at 37 °C under vigorous shaking.

**Reporting summary.** Further information on research design is available in the Nature Research Reporting Summary linked to this article.

## Data availability

Source data are provided with this paper.

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

## Acknowledgements

We thank Mona Orr for technical help. We thank José Lemos for the gift of pGpp standard. This work is supported, in part, by an R35 GM127088 from NIGMS (to J.D.W.), a GRFP DGE-1256259 from NSF (to B.W.A.), 1715710 from NSF (to D.A.-N.), and R01AI110740 from NIAID and NIDDK (to V.T.L.).

## Author contributions

B.W.A., V.T.L., and J.D.W. conceptualized the DRaCALA screen experiments. J.Y., B.W.A., A.T., H.T., V.T.L., and J.D.W. performed the library construction, J.Y., B.W.A., A.T., and H.T. performed the DRaCALA screen experiments. J.Y. and J.D.W. conceptualized the experiments on NahA. J.Y. performed most biochemical and physiological experiments. J.Y., D.M.S., and D.A.-N. designed LC-MS experiments. J.Y. and D.M.S. performed LC-MS experiments. J.Y. analyzed data for this work. J.Y. and J.D.W. wrote the manuscript. All authors edited the manuscript. V.T.L. and J.D.W. supervised the study. Funding was acquired by J.D.W. and V.T.L.

## Competing interests

The authors declare no competing interests.
