## [Peer Review File · Nature Communications]

REVIEWER COMMENTS

Reviewer #1 (Remarks to the Author):

This work is very important because it is the first plausible proposal explaining how *Bacillus* species might solve the dilemma raised because lowering of GTP pool levels leads to negative regulation of rRNA transcription initiation while simultaneously modulating translational activity on one hand and purine biosynthetic activity on the other. The answer arises from finding altered patterns of pppGpp, ppGpp and pGpp binding with genomic arrays of purified proteins. These changes are mediated by a Nudix enzyme, termed NahA (*subtilis*) / Yvcl (*anthracis*). One group of six enzymes within the de novo purine pathway can be regulated by binding of any of the three alarmones. A separate group of enzymes that function in translation and GTPase-mediated ribosome assembly are found to bind pppGpp and ppGpp but not pGpp. Enrichment of pGpp levels at the expense of pppGpp, ppGpp or both would therefore preferentially inhibit de novo purine synthesis. This led to finding that the *B. subtilis* nahA NuDiX hydrolase should enrich pGpp at the expense of pppGpp and ppGpp made during the stringent response.

Critique

1. The authors convincingly verify catalytic specificity expectations for the NahA enzyme in vitro as well as the in vivo predictions of deleting NahA on differential abundance of alarmone pool levels using HPLC, TLC and ms. DRaCALA assays show all three alarmones bind to purine nucleotide synthesis enzymes (HprT, PurA, Xpt, GuaC, and Gmk) and confirm the expected binding hierarchy of pGpp > ppGpp > pppGpp. Conversely, ribosome and translation regulation proteins HflX, Obg, Der, RbgA and Era do not bind pGpp but do bind ppGpp and pppGpp with higher affinities for the tetraphosphate got HflX, Obg and Der (Fig 3).

It is shown that enzymatic hydrolysis of pGpp can occur using the same enzyme as for (p)ppGpp, namely by the Mn⁺⁺-dependent hydrolase activity of the bifunctional *B. subtilis* RSH protein, sometimes called Rel, but not the term RelA used by the authors. Observed initial rates of removal of the 3'-pyrophosphate moiety are even faster for pGpp than for (p)ppGpp.

2 NahA deletion-dependent changes of relative alarmone abundance during growth and in stationary phase were found consistent with expectations and complemented by NahA complementation. Comparisons during arginine hydroxamate-induced alarmone accumulation surprisingly revealed some pGpp in the nahA deleted strain, which the authors attribute to a small alarmone synthetase (SAS1) activity, which is independently verified..

The effects of deleting nahA on metabolic pools of selected nucleobases, nucleoside and nucleotides during mid-log and stationary phase growth were presented Fig 4A. I assume, although not stated, that the media does not contain with nucleosides or bases. The values for the three alarmones serve as internal controls. The absence of pGpp confers few effects on common ribonucleoside mono-, di- and triphosphate abundance. In mid-log guanine, but not adenine, abundance increases without pGpp and decreases in stationary phase. In stationary phase guanine abundance drops without pGpp and with adenine less responsive. Since these strains are spore formers it is hard for me to interpret this behavior solely in terms of purine synthesis effects. The major changes seem relegated to the distribution of the three alarmones.

3 The task then turns to showing preferential effects on protein modulation. This is addressed by effects of pGpp on protein translation as judged by rates of protein 35S-labeling during a stringent response induced by arginine hydroxamate addition due to blocked arginine tRNA charging. The observed inhibitory effects are modest as are the effects on growth whether deleting or complemented by inducing NahA. Since pGpp does not bind to translation component proteins, one might expect alterations of translational modulation to persist as for the deletion... the result observed here. I believe inducing stringent response in a different way might allow observing otherwise hidden modulation differences. It has been argued for *E. coli* that the use of AA-hydroxamates to induce the stringent response puts a choke hold on the ability to modulate

translation via several mechanisms. First, slowing of protein synthesis during simple starvation reduces consumption of charged tRNA, which can modulate the stringent response. A second is that simple starvation leads to rapid turnover of tRNA. A third is that ribosomes are degraded and assembly is altered during simple starvation.

4. A modest lag for the mutant during outgrowth after downshift is noticed for the deletion that is complemented by inducing NahA. I would like to note that the interpretation from a linear plot of lag time differences before exponential growth is established is difficult to interpret. A semi-log plot is more typical. This reader, at least, wonders whether given this result, whether downshifts in media containing purines would be different.

5 The authors make the assumption that binding of all alarmones is linearly related to inhibition of catalytic activity. Just to be extra careful, might it be worth it to look at a few competition for a few kinetic examples. There are exceptions to this rule. Adenylosuccinate synthetase is GTP-activated and inhibited by ppGpp binding to the GTP site which transmits serial conformational changes across the protein to the adenylosuccinate catalytic site (Hou Z et al., 1999; PMID: 10364182). For EF-Tu ppGpp was found to bind more tightly than GDP, which made us think ppGpp inhibition of protein synthesis might occur by blocking this step in protein synthesis. However EF-Ts mediated exchange of bound ppGpp for GTP was unaffected despite tighter binding of ppGpp than GDP (Miller et al 1973. PMID: 4570843).

Mike Cashel

Reviewer #2 (Remarks to the Author):

In the present manuscript, Yang et al. performed extensive screens employing a *Bacillus anthracis* ORF library to characterize and compare the interactomes of the second messengers pppGpp, ppGpp and pGpp. While ppGpp and pppGpp (collectively denoted as (p)ppGpp) are rather well-studied, the functional role of pGpp (and its underlying target proteins) is widely unaddressed. By their screens the authors uncover – in agreement with similar studies done for *Staphylococcus aureus* and *Escherichia coli* - that ppGpp and pppGpp primarily interact with proteins involved in nucleotide metabolism and ribosome biogenesis/translation. pGpp, in contrast, only affects the proteins of the former group thus evidencing differences between the pGpp and (p)ppGpp species. The authors furthermore identified a hydrolase (YvcI/NahA) that converts both ppGpp and pppGpp to pGpp a function apparently important in the outgrowth after a nutritional stress of *B. subtilis*. This is the first work that addresses the interactome of the pGpp nucleotide in great detail, compares it to (p)ppGpp and furthermore demonstrates their regulatory disparity in vivo. The study will probably initiate future work on the regulatory role of pGpp. This, in my opinion, merits publication in *Nature Communications*.

I do not advise for further experiments, because the experiments currently presented are properly conducted and legitimate the conclusions of the manuscript. There are some ambiguities mainly in the presentation of the DRaCALA results, which must be addressed by the authors before publication.

- Inconsistencies in presentation of the DRaCALA results between figure 1b and table S1

1) Table S1 denotes scores for pGpp/ppGpp/pppGpp binding to selected proteins of the ORF libraries, while figure 1b gives an overview of their target spectra. It is however unclear to me what exactly is depicted in figure 1b. The authors conducted their DRaCALA screen with libraries overexpressing His-tagged or His-MBP-tagged proteins (figure 1a). Table S1 presents data for pGpp/ppGpp/pppGpp binding to the His-MBP library but for the His library on binding of pppGpp. Where are the pGpp and ppGpp data?! And is the overview in figure 1b based on the His-MBP

proteins? That should be stated in the figure legend.

This is important because there are some target proteins (i.e. YvcI/NahA, BA4322, BA3046, BA3026 and BA3392) where the results are different between the His-MBP and His library.

2) Also, some data are in the table in columns G and H referring to a ,pppGpp-GTP screen'. Potentially, this one was conducted with ³²P-labelled pppGpp in presence of unlabelled GTP to reduce background through binding of pppGpp to GTPases apart (competed by the excess of GTP). But I find no mention in the main text or the materials&methods section detailing those data. The rationale for those data (His-MBP versus His; ,pppGpp-GTP screen') and their interpretation should be provided in the main text.

Minor comments to figure 1b and table S1: Please denote the 'BA'-locus for the proteins in 1b; the names in the figure 1b and table S1 are not consistent everywhere, e.g. (p)ppGpp synthetase vs. GTP pyrophosphokinase RelA, there is only 1x 5' nucleotidase in fig. 1b but 2x in the table. MutT and YvcI are both Nudix hydrolases so it is unclear to which one is referred to in 1b.

- Some methods are not extensively described in the manuscript, i.e.

1) Growth recovery from nutrient downshift (ll. 533-537) very poorly described; What is the composition of the media (amount of amino acids and/or reference)? What OD exactly is meant with the 'mid-log phase'? To what OD were the freshly diluted cultures adjusted? There is also no mention of the inducer IPTG here....

2) Overexpression and purification of NahA (ll. 405 onwards): I can not find the procedure for the His-MBP-tagged YvcI, which appears in SI figure 2.

3) I could not find a protocol for the lysis of the ORF library (mentioned in l. 360). Please provide description and/or reference.

- Typos or related

Reviewer #3 (Remarks to the Author):

Yang et al performed a comprehensive screening of the binding proteins of pppGpp and ppGpp in Bacillus. They found proteins in purine nucleotide synthesis and translational GTPases as shared targets of pppGpp and ppGpp, which confirmed the conserved targets of (p)ppGpp in both E. coli (proteobacteria) and Bacillus (Staphylococcus, Firmicutes). Interestingly, the authors also found a Nudix hydrolase YvcI (NahA), which despite similarities in both sequence and the ability to cleave (p)ppGpp to the E. coli Nudix enzymes NudG, MutT, actually produces pGpp from (p)ppGpp, which the authors confirmed by several methods. More importantly, via a series of experiments the authors confirmed that the constitutively expressed NahA produces pGpp from (p)ppGpp in vivo in Bacillus, which serves to reduce the level of (p)ppGpp upon stresses. Moreover, a third comprehensive screening of the pGpp binding proteins revealed that pGpp selectively targets purine nucleotide synthesis enzymes but not the translational GTPases. These and further data led the authors to the idea that NahA fine-tunes the alarmone composition upon stress by depleting (p)ppGpp to de-repress translation and producing pGpp to maintain the suppression of purine synthesis, which together contribute to fast regrowth fitness. Given the solid evidence of the presence of pGpp in Bacillus and its distinct binding targets, pGpp shall formally be considered the third alarmone besides (p)ppGpp in Gram-positive bacteria.

The experimental techniques and analysis methods are solid. The data presented are strong and convincing, and the manuscript is concise and clear.

1. L125-126, the negative data of YvcI with GTP and 8-OXO-GTP may be still important to show, in a supplement.

Some other minor errors to check throughout the manuscript

2. L367, missing ref.

3. L406, *E. coli* not italic.

4. L521, Missing units

By Yong Zhang

RESPONSE TO REVIEWER COMMENTS

Reviewer #1 (Remarks to the Author):

This work is very important because it is the first plausible proposal explaining how *Bacillus* species might solve the dilemma raised because lowering of GTP pool levels leads to negative regulation of rRNA transcription initiation while simultaneously modulating translational activity on one hand and purine biosynthetic activity on the other. The answer arises from finding altered patterns of pppGpp, ppGpp and pGpp binding with genomic arrays of purified proteins. These changes are mediated by a Nudix enzyme, termed NahA (*subtilis*) / Yvcl (*anthracis*). One group of six enzymes within the de novo purine pathway can be regulated by binding of any of the three alarmones. A separate group of enzymes that function in translation and GTPase-mediated ribosome assembly are found to bind pppGpp and ppGpp but not pGpp. Enrichment of pGpp levels at the expense of pppGpp, ppGpp or both would therefore preferentially inhibit de novo purine synthesis. This led to finding that the *B. subtilis* nahA NuDiX hydrolase should enrich pGpp at the expense of pppGpp and ppGpp made during the stringent response.

Response: Thanks! We appreciate your comment that the work explained how Bacillus solve the dilemma of how ppGpp regulates both translational activity and purine biosynthesis by altered patterns of pppGpp, ppGpp and pGpp binding of proteins.

Critique

1. The authors convincingly verify catalytic specificity expectations for the NahA enzyme in vitro as well as the in vivo predictions of deleting NahA on differential abundance of alarmone pool levels using HPLC, TLC and ms. DRaCALA assays show all three alarmones bind to purine nucleotide synthesis enzymes (HprT, PurA, Xpt, GuaC, and Gmk) and confirm the expected binding hierarchy of pGpp > ppGpp > pppGpp. Conversely, ribosome and translation regulation proteins HflX, Obg, Der, RbgA and Era do not bind pGpp but do bind ppGpp and pppGpp with higher affinities for the tetraphosphate got HflX, Obg and Der (Fig 3).

It is shown that enzymatic hydrolysis of pGpp can occur using the same enzyme as for (p)ppGpp, namely by the Mn⁺⁺-dependent hydrolase activity of the bifunctional *B. subtilis* RSH protein, sometimes called Rel, but not the term RelA used by the authors. Observed initial rates of removal of the 3'-pyrophosphate moiety are even faster for pGpp than for (p)ppGpp.

Response: We changed the name of the B. subtilis RSH protein to Rel in our manuscript. However, because B. subtilis RelA has been identified and the nomenclature widely used for 30 years in B. subtilis papers, we also keep the name "RelA" for consistency with prior literature.

2 NahA deletion-dependent changes of relative alarmone abundance during growth and in stationary phase were found consistent with expectations and complemented by NahA complementation. Comparisons during arginine hydroxamate-induced alarmone accumulation surprisingly revealed some pGpp in the nahA deleted strain, which the

authors attribute to a small alarmone synthetase (SAS1) activity, which is independently verified.

The effects of deleting nahA on metabolic pools of selected nucleobases, nucleoside and nucleotides during mid-log and stationary phase growth were presented Fig 4A. I assume, although not stated, that the media does not contain with nucleosides or bases.

Response: Yes, the media we used for LC-MS analysis did not contain nucleosides or bases. We clarify this in the revision. "Unless otherwise indicated, B. subtilis strains were grown in S7 defined media with modifications (50 mM MOPS instead of 100 mM, 0.1% potassium glutamate, 1% glucose, with no additional amino acids, nucleosides or nucleobases), with shaking at 250 rpm 37 °C."

The values for the three alarmones serve as internal controls. The absence of pGpp confers few effects on common ribonucleoside mono-, di- and triphosphate abundance. In mid-log guanine, but not adenine, abundance increases without pGpp and decreases in stationary phase. In stationary phase guanine abundance drops without pGpp and with adenine less responsive. Since these strains are spore formers it is hard for me to interpret this behavior solely in terms of purine synthesis effects. The major changes seem relegated to the distribution of the three alarmones.

Response: Yes, the strains are spore formers. The time scale of our LC-MS experiments (less than 4 hours starting from mid exponential phase, the last time points are long before the cells have fully reached stationary phase) are too short for the formation of spores.

3 The task then turns to showing preferential effects on protein modulation. This is addressed by effects of pGpp on protein translation as judged by rates of protein ³⁵S-labeling during a stringent response induced by arginine hydroxamate addition due to blocked arginine tRNA charging. The observed inhibitory effects are modest as are the effects on growth whether deleting or complemented by inducing NahA. Since pGpp does not bind to translation component proteins, one might expect alterations of translational modulation to persist as for the deletion... the result observed here. I believe inducing stringent response in a different way might allow observing otherwise hidden modulation differences. It has been argued for E. coli that the use of AA-hydroxamates to induce the stringent response puts a choke hold on the ability to modulate translation via several mechanisms. First, slowing of protein synthesis during simple starvation reduces consumption of charged tRNA, which can modulate the stringent response. A second is that simple starvation leads to rapid turnover of tRNA. A third is that ribosomes are degraded and assembly is altered during simple starvation.

Response: Good suggestion. In the revision, we performed ³⁵S-incorporation assay with simple amino acid starvation to avoid potential complicated effect of arginine hydroxamate. This new assay showed stronger effect of nahA deletion on protein synthesis. This new result updated the Figure 4b and the corresponding text. "We induce amino acid starvation either by resuspending cells from amino acid replete medium to minimal medium or by adding the nonfunctional amino acid analog arginine hydroxamate. In both cases, the rate of ³⁵S-methionine incorporation is significantly lower in ΔnahA than in wild type, indicating a stronger inhibition of translation (Figure 4b, Figure S6a)."

4. A modest lag for the mutant during outgrowth after downshift is noticed for the deletion that is complemented by inducing NahA. I would like to note that the interpretation from a linear plot of lag time differences before exponential growth is established is difficult to interpret. A semi-log plot is more typical. This reader, at least, wonders whether given this result, whether downshifts in media containing purines would be different.

Response: We replotted the figure in log scale (Figure 4e), which confirmed our observation of a lag time difference. In addition, we performed downshift recovery experiment with guanosine presence and found out that all strains had the same lag phase length during their recovery in rich media (Figure S6c).

5 The authors make the assumption that binding of all alarmones is linearly related to inhibition of catalytic activity. Just to be extra careful, might it be worth it to look at a few competition for a few kinetic examples. There are exceptions to this rule. Adenylosuccinate synthetase is GTP-activated and inhibited by ppGpp binding to the GTP site which transmits serial conformational changes across the protein to the adenylosuccinate catalytic site (Hou Z et al., 1999; PMID: 10364182). For EF-Tu ppGpp was found to bind more tightly than GDP, which made us think ppGpp inhibition of protein synthesis might occur by blocking this step-in protein synthesis. However, EF-Ts mediated exchange of bound ppGpp for GTP was unaffected despite tighter binding of ppGpp than GDP (Miller et al 1973. PMID: 4570843).

Response: This is a good point. We have added the reviewer's comment in the manuscript as a precaution. For the targets in purine biosynthesis pathway, we present data showing that pGpp inhibits HprT stronger than (p)ppGpp (Figure 3e). We also examined pGpp inhibition of Xpt and found that it inhibits Xpt more strongly than (p)ppGpp, correlating with its higher affinity (recently published Anderson et al. 2020). Corresponding contents are now reflected in the main text.

Mike Cashel

Reviewer #2 (Remarks to the Author):

In the present manuscript, Yang et al. performed extensive screens employing a *Bacillus anthracis* ORF library to characterize and compare the interactomes of the second messengers pppGpp, ppGpp and pGpp. While ppGpp and pppGpp (collectively denoted as (p)ppGpp) are rather well-studied, the functional role of pGpp (and its underlying target proteins) is widely unaddressed. By their screens the authors uncover – in agreement with similar studies done for *Staphylococcus aureus* and *Escherichia coli* – that ppGpp and pppGpp primarily interact with proteins involved in nucleotide metabolism and ribosome biogenesis/translation. pGpp, in contrast, only affects the proteins of the former group thus evidencing differences between the pGpp and (p)ppGpp species. The authors furthermore identified a hydrolase (Yvcl/NahA) that converts both ppGpp and pppGpp to pGpp a function apparently important in the

outgrowth after a nutritional stress of *B. subtilis*.

This is the first work that addresses the interactome of the pGpp nucleotide in great detail, compares it to (p)ppGpp and furthermore demonstrates their regulatory disparity *in vivo*. The study will probably initiate future work on the regulatory role of pGpp. This, in my opinion, merits publication in Nature Communications.

I do not advice for further experiments, because the experiments currently presented are properly conducted and legitimate the conclusions of the manuscript. There are some ambiguities mainly in the presentation of the DRaCALA results, which must be addressed by the authors before publication.

- Inconsistencies in presentation of the DRaCALA results between figure 1b and table S1

1) Table S1 denotes scores for pGpp/ppGpp/pppGpp binding to selected proteins of the ORF libraries, while figure 1b gives an overview of their target spectra. It is however unclear to me what exactly is depicted in figure 1b. The authors conducted their DRaCALA screen with libraries overexpressing His-tagged or His-MBP-tagged proteins (figure 1a). Table S1 presents data for pGpp/ppGpp/pppGpp binding to the His-MBP library but for the His library on binding of pppGpp. Where are the pGpp and ppGpp data?! And is the overview in figure 1b based on the His-MBP proteins? That should be states in the figure legend.

This is important because there are some target proteins (i.e. Yvcl/NahA, BA4322, BA3046, BA3026 and BA3392) where the results are different between the His-MBP and His library.

Response: Thank you for pointing out the lack of clarity. For clarification, pppGpp results were obtained using both His-tagged and His-MBP-tagged proteins. ppGpp and pGpp results were obtained with only His-MBP-tagged proteins. This statement has been added to the legend of Figure 1b.

The differences are due to use of different libraries- differential expression levels as well as the difference introduced by tagging, different solubility of His-tagged and HisMBP-tagged proteins (MBP tends to increase the solubility of tagged protein). In general, when the results disagree, they are more likely due to false negatives in one of the screens. Therefore, by doing screens with two different tagging methods, we can maximize identification of correct targets.

2) Also, some data are in the table in columns G and H referring to a 'pppGpp-GTP screen'. Potentially, this one was conducted with ³²P-labelled pppGpp in presence of unlabelled GTP to reduce background through binding of pppGpp to GTPases apart (competed by the excess of GTP). But I find no mention in the main text or the materials&methods section detailing those data.

The rationale for those data (His-MBP versus His; ,pppGpp-GTP screen') and their interpretation should be provided in the main text.

Response: The reviewer is correct that 'pppGpp-GTP screen' does refer to DRaCALA with ³²P-labeled pppGpp in the presence of excessive unlabeled GTP. We should not have put these data in this manuscript- they were performed for a different purpose than for this work. Because this dataset is not relevant to this manuscript, we deleted this dataset from this manuscript.

Minor comments to figure 1b and table S1: Please denote the 'BA'-locus for the proteins in 1b; the names in the figure 1b and table S1 are not consistent everywhere, e.g. (p)ppGpp synthetase vs. GTP pyrophosphokinase RelA, there is only 1x 5' nucleotidase in fig. 1b but 2x in the table. MutT and Yvcl are both Nudix hydrolases so it is unclear to which one is referred to in 1b.

Response: We added the 'BA'-locus tags for identified B. anthracis targets for reference. The nomenclature of gene names are unified between Figure 1b and Table S1; 5'-nucleotidases and NuDiX hydrolases are now labeled with 'BA'-locus tags to differentiate them.

• Some methods are not extensively described in the manuscript, i.e.

1) Growth recovery from nutrient downshift (II. 533-537) very poorly described; What is the composition of the media (amount of amino acids and/or reference)? What OD exactly is meant with the 'mid-log phase'? To what OD were the freshly diluted cultures adjusted? There is also no mention of the inducer IPTG here....

Response: Media components were originally listed in the "Plasmid and Strain Construction and Growth Conditions" section. Now the corresponding statement has been added in the "Growth Recovery from Nutrient Downshift" result section: Cells were grown in defined amino acid depleted medium (S7 medium with glucose, 20 amino acids and 0.5 mM IPTG) to mid log phase ($OD_{600nm} \sim 0.5$), and then washed in S7 glucose without amino acid). Washed cultures were diluted into fresh medium with 20 amino acids and 0.5 mM IPTG to $OD_{600nm} = 0.01$ and the growth was monitored by a plate reader (Synergy 2, Biotek) at 37 °C under vigorous shaking."

2) Overexpression and purification of NahA (II. 405 onwards): I can not find the procedure for the His-MBP-tagged Yvcl, which appears in SI figure 2.

Response: The procedure for His-MBP-Yvcl(NahA) purification has been added in the method section "Overexpression and Purification of NahA." "His-MBP-NahA was prepared using similar protocol as mentioned above with modifications: pVL847-nahA was transformed into E. coli and all the media contain 10 µg/mL gentamycin instead of kanamycin; SUMO-protease and His-SUMO tag removal were not applied."

3) I could not find a protocol for the lysis of the ORF library (mentioned in I. 360). Please provide description and/or reference.

Response: The protocol for the lysis of the ORF library has been added at the end of the section "Bacillus anthracis ORFeome Library Construction." E. coli strains containing overexpression vectors were grown in LB-M9 media (7 g/L Na_2HPO_4 , 2 g/L KH_2PO_4 , 0.5 g/L NaCl, 1 g/L NH_4Cl , 2 g/L glucose, 1 g/L sodium succinate dibasic hexahydrate; 10 g/L tryptone, 5 g/L yeast extract and 3 mM $MgSO_4$) at 30 °C and the overexpression was induced by 0.5 mM IPTG. Cells were collected by centrifugation at 5000 g 4 °C for 10 min. Cells were resuspended in 10% original volume of resuspension/lysis buffer (10 mM Tris pH 7.5, 100 mM NaCl, 5 mM $MgCl_2$, 0.25 mg/mL lysozyme, 0.01 mg/mL DNase I, 1 mM

PMSF). Cells were then lysed by 3 cycles of -80 °C freezing and room temperature thawing.

Reviewer #3 (Remarks to the Author):

Yang et al performed a comprehensive screening of the binding proteins of pppGpp and ppGpp in Bacillus. They found proteins in purine nucleotide synthesis and translational GTPases as shared targets of pppGpp and ppGpp, which confirmed the conserved targets of (p)ppGpp in both E. coli (proteobacteria) and Bacillus (Staphylococcus, Firmicutes). Interestingly, the authors also found a Nudix hydrolase Yvcl (NahA), which despite similarities in both sequence and the ability to cleave (p)ppGpp to the E. coli Nudix enzymes NudG, MutT, actually produces pGpp from (p)ppGpp, which the authors confirmed by several methods. More importantly, via a series of experiments the authors confirmed that the constitutively expressed NahA produces pGpp from (p)ppGpp in vivo in Bacillus, which serves to reduce the level of (p)ppGpp upon stresses. Moreover, a third comprehensive screening of the pGpp binding proteins revealed that pGpp selectively targets purine nucleotide synthesis enzymes but not the translational GTPases. These and further data led the authors to the idea that NahA fine-tunes the alarmone composition upon stress by depleting (p)ppGpp to de-repress translation and producing pGpp to maintain the suppression of purine synthesis, which together contribute to fast regrowth fitness. Given the solid evidence of the presence of pGpp in Bacillus and its distinct binding targets, pGpp shall formally be considered the third alarmone besides (p)ppGpp in Gram-positive bacteria. The experimental techniques and analysis methods are solid. The data presented are strong and convincing, and the manuscript is concise and clear.

1. L125-126, the negative data of Yvcl with GTP and 8-OXO-GTP may be still important to show, in a supplement.

Response: The negative data of GTP is in Figure S3a; the negative data of 8-oxo-GTP is added as Figure S3c. Corresponding contents are updated in the main text: "Yvcl failed to hydrolyze either GTP (Figure S3a) or 8-oxo-GTP (Figure S3c)."

Some other minor errors to check throughout the manuscript

2. L367, missing ref.

Response: The missing reference has been cited: Burby, P. & Simmons, L. CRISPR/Cas9 Editing of the Bacillus subtilis Genome. BIO-PROTOCOL 7, (2017).

3. L406, E. coli not italic.

Response: The format has been fixed.

4. L521, Missing units

Response: We checked the PDF file and the original manuscript and found out that the formatting error led to the loss of "°C" label. Now the issue has been fixed.

By Yong Zhang

REVIEWERS' COMMENTS

Reviewer #1 (Remarks to the Author):

This work is important because it is the first plausible proposal explaining how *Bacillus* species might solve the dilemma raised because lowering of GTP pool levels leads to negative regulation of rRNA transcription initiation while simultaneously modulating translational activity on one hand and purine biosynthetic activity on the other. The answer arises from finding altered patterns of pppGpp, ppGpp and pGpp binding with genomic arrays of purified proteins. These changes are mediated by a Nudix enzyme, termed NahA (*subtilis*) / Yvcl (*anthracis*). One group of six enzymes within the de novo purine pathway can be regulated by binding of any of the three alarmones. A separate group of enzymes that function in translation and GTPase-mediated ribosome assembly are found to bind pppGpp and ppGpp but not pGpp. Enrichment of pGpp levels at the expense of pppGpp, ppGpp or both would therefore preferentially inhibit de novo purine synthesis. This led to finding that the *B. subtilis* nahA NuDiX hydrolase should enrich pGpp at the expense of pppGpp and ppGpp made during the stringent response.

The authors convincingly verify catalytic specificity expectations for the NahA enzyme in vitro as well as the in vivo predictions of deleting NahA on differential abundance of alarmone pool levels using HPLC, TLC and ms. DRaCALA assays show all three alarmones bind to purine nucleotide synthesis enzymes (HprT, PurA, Xpt, GuaC, and Gmk) and confirm the expected binding hierarchy of pGpp > ppGpp > pppGpp. Conversely, ribosome and translation regulation proteins HflX, Obg, Der, RbgA and Era do not bind pGpp but do bind ppGpp and pppGpp with higher affinities for the tetraphosphate got HflX, Obg and Der (Fig 3).

It is shown that enzymatic hydrolysis of pGpp can occur using the same enzyme as for (p)ppGpp, namely by the Mn⁺⁺-dependent hydrolase activity of the bifunctional *B. subtilis* RSH protein, sometimes called Rel, but not the term RelA used by the authors. Observed initial rates of removal of the 3'-pyrophosphate moiety are even faster for pGpp than for (p)ppGpp. NahA deletion-dependent changes of relative alarmone abundance during growth and in stationary phase were found consistent with expectations and complemented by NahA complementation. Comparisons during arginine hydroxamate-induced alarmone accumulation surprisingly revealed some pGpp in the nahA deleted strain, which the authors attribute to a small alarmone synthetase (SAS1) activity, which is independently verified..

The effects of deleting nahA on metabolic pools of selected nucleobases, nucleoside and nucleotides during mid-log and stationary phase growth were presented Fig 4A. I assume, although not stated, that the media does not contain with nucleosides or bases. The values for the three alarmones serve as internal controls. The absence of pGpp confers few effects on common ribonucleoside mono-, di- and triphosphate abundance. In mid-log guanine, but not adenine, abundance increases without pGpp and decreases in stationary phase. In stationary phase guanine abundance drops without pGpp and with adenine less responsive. Since these strains are spore formers it is hard for me to interpret this behavior solely in terms of purine synthesis effects. The major changes seem relegated to the distribution of the three alarmones. The task then turns to showing preferential effects on protein modulation

This is addressed by effects of pGpp on protein translation as judged by rates of protein 35S-labeling during a stringent response induced by arginine hydroxamate addition due to blocked arginine tRNA charging. The observed inhibitory effects are modest as are the effects on growth whether deleting or complemented by inducing NahA. Since pGpp does not bind to translation component proteins, one might expect alterations of translational modulation to persist as for the deletion... the result observed here. I believe inducing stringent response in a different way might allow observing otherwise hidden modulation differences. It has been argued for *E. coli* that the use of AA-hydroxamates to induce the stringent response puts a choke hold on the ability to modulate translation via several mechanisms. First, slowing of protein synthesis during simple starvation reduces consumption of charged tRNA, which can modulate the stringent response. A second is that simple starvation leads to rapid turnover of tRNA. A third is that ribosomes are degraded and assembly is altered during simple starvation.

A modest lag for the mutant during outgrowth after downshift is noticed for the deletion that is complemented by inducing NahA. I would like to note that the interpretation from a linear plot of lag time differences before exponential growth is established, is very difficult to interpret in terms of when exponential growth starts. A semi-log plot is more useful. This reader, at least, wonders if given this result, whether downshifts in media containing purines would be different.

Reviewer #2 (Remarks to the Author):

The authors have addressed all my points. I congratulate them to this lovely piece of work.